



# Automatic detection and correction of pitch misalignment in wind turbine rotors

Marta Bertelè[1], Carlo L. Bottasso[1,2], and Stefano Cacciola[2]

[1]Wind Energy Institute, Technische Universität München, Garching bei München, D-85748 Germany,
[2]Dipartimento di Scienze e Tecnologie Aerospaziali, Politecnico di Milano, Milano, I-20156 Italy

**Correspondence:** C.L. Bottasso (carlo.bottasso@tum.de)

**Abstract.**

In this work, a new algorithm is presented to correct for pitch misalignment imbalances of wind turbine rotors. The method uses signals measured in the fixed frame of the machine, typically in the form of accelerations or loads. The amplitude of the one per revolution signal harmonic is used to quantify the imbalance, while its phase to locate the unbalanced blade(s).

The near linearity of the unknown relationship between harmonic amplitude and pitch misalignment is used to derive a simple algorithm that iteratively rebalances the rotor. This operation is conducted while the machine is in operation, without the need of shutting it down. The method is not only applicable to the case of a single misaligned blade, but also to the generic case of multiple concurrent imbalances. A part from the availability of acceleration or load sensors, the method requires the ability of the rotor blades to be commanded independently from one another, which is typically possible on many modern machines. The

new method is demonstrated in a realistic simulation environment, using an aeroservoelastic wind turbine model in a variety of wind and operating conditions.

## 1 Introduction

The pitch system has the highest failure rate of all wind turbine components (Wilkinson et al., 2010). Issues can include, among others, faults of the pitch actuators or of the pitch angle sensors, but they can also be caused by an imperfect installation of

the blades. In general, rotor asymmetries represent a significant problem for wind turbines, as also witnessed by the fact that certification guidelines require for the verification of the effects of even relatively small pitch misalignments (typically $\pm0.3°$ for two blades, cf. GL Standards (2010), §4.3.4.1 pp.4–20).

Irrespectively of the specific type of fault, a pitch imbalance will have as a direct consequence not only a possible decrease in harvested energy but, most importantly, an increased level of vibrations and rotor speed fluctuations (Kusiak and Verma, 2011;

Hyers et al., 2006). In fact, when a pitch misalignment among the blades is present, the periodic aerodynamic, dynamic and gravitational loading experienced by the blades is not balanced. As a result, additional harmonic components are transferred from the rotating to the fixed frame, resulting in vibrations that may lead to the failure of other components of the machine, but that may also affect its fatigue life if not promptly corrected for (Yang et al., 2008). Moreover, whenever vibrations are fed back to the turbine control laws, imbalances can also result in increased duty cycles for the machine actuators.





Currently, the downtime related to pitch failures is relatively high (Wilkinson et al., 2010). In fact, once an anomalous behavior has been detected —typically by higher than expected fixed-frame vibrations, see Hameed et al. (2009)— pitch correction operations are often initiated by a visual inspection. An operator (more recently with the possible aid of a drone) takes pictures or videos of the blades, which are later analyzed to reveal whether all blades have the same pitch angle. Once a pitch offset has been estimated, the blade pitch is reset to align it with the others. This operation will imply some downtime and may come at a non-negligible cost. Furthermore, the procedure might not always be able to produce an exactly balanced rotor. Clearly, more effective condition monitoring and correction strategies for the pitch system of wind turbines should be developed. An ideal solution should be able to first identify when a rotor is unbalanced, and then to automatically rebalance it. This should be obtained without the need to shut down the machine, without the presence and supervision of an operator, and without the need for expensive extra hardware.

Imbalance detection techniques have been developed in the literature, with the goal of providing operators with an improved knowledge on the status of their machines than the simple boolean indication "balanced/unbalanced" turbine. For example, Niebsch et al. (2010) and Niebsch and Ramlau (2014) proposed a method to simultaneously estimate from nacelle vibrational measurements both mass and aerodynamic imbalance effects. The method considers a finite element model of the turbine, and the imbalance terms are obtained by solving an inverse problem through nonlinear regularization theory. Results are interesting although not excellent, with errors in the estimation of the pitch misalignment up to $0.5°$. However, the need for a detailed model of the machine may hinder the applicability of this method. A different approach has been proposed by Kusnick et al. (2015). In this case, the blade misalignment estimation is performed by an ad hoc work-flow using multiple measurements, including power output, blade loads and accelerations. Finally, a method based on system identification is presented by Cacciola et al. (2016). In that work, a neural network is trained based on nodding moment and power measurements from different experiments conducted for varying known pitch misalignments and operating conditions. After training, the network is able to detect the severity and location of the imbalance, even distinguishing effects caused by pitch misalignments from those induced by ice accretion.

Ad-hoc controllers have also been formulated to correct for rotor imbalances (Kanev and van Engelen, 2009; Kanev et al., 2009; Petrović et al., 2015; Cacciola and Riboldi, 2017; Cacciola et al., 2017). In all theses cases, the general idea is to develop a control law that compensates for a pitch misalignment by targeting imbalance-induced vibrations, typically by Coleman-transforming blade loads (Bossanyi, 2003). One possible drawback of such approaches is that any bias in the calibration of load sensors may result in an erroneous pitch misalignment compensation, as it might be hard —or altogether impossible— to distinguish between a calibration error and an imbalance-born blade load.

The analysis of signals such as loads and accelerations measured on the wind turbine fixed frame provides for a way to identify if a rotor is unbalanced. In fact, it is well known that the amplitude of the 1P (once per revolution) harmonic is an indicator of an unbalanced rotor. Recently, it was shown that the phase of that same harmonic can be used to identify the unbalanced blade(s) (Cacciola et al., 2016). Based on this simple signal analysis, a condition monitoring system can be developed to detect severity and location of the imbalance, in order to schedule appropriate maintenance and repair actions.



In the present work, the same concept is used to automatically rebalance an unbalanced rotor. In a nutshell, the method works as follows. First, an unknown linear relationship is assumed between pitch setting of the blades and 1P amplitude of a signal measured in the fixed frame. Exploiting the radial symmetry of a rotor, the model coefficients are reduced to only two. In addition, this has also the effect of including the phase information in the model, which eventually allows one to correctly

identify the pitch misalignment of each blade. Since the linear imbalance-disturbance model is determined by two parameters, one single additional measurement (in addition to the one performed on the currently unbalanced configuration) is necessary to identify the unknown imbalance-disturbance relationship. This is easily achieved by pitching the blades by some amount and measuring the resulting 1P amplitude. Once the linear relationship is known, it is trivial to compute the blade pitch offset that, by zeroing the 1P amplitude, balances the rotor. To account for possible small non-linearities, the procedure can be iterated

a few times, as necessary. A similar approach was presented in Bertelè et al. (2017), which nevertheless considered only the case of a pitch fault located in one single blade. The present work expands and generalizes this methodology, allowing for the detection and correction of multiple simultaneous pitch imbalances.

The paper is organized as follows. Section 2 formulates the method used for the proposed imbalance detection and correction procedure. In particular, §2.1 shows the mechanism through which a pitch imbalance causes a 1P load in the fixed frame, by

developing a spectral analysis of the relevant loads and explaining their origin. Next, §2.2 formulates the linear imbalance-disturbance model of an axial-symmetric rotor, while §2.3 shows how the model coefficients can be readily identified by using two fixed frame measurements at two different pitch settings. Lastly, §2.4 explains the rebalancing procedure. Results are discussed in section 3, which reports extensive numerical simulations performed with a state-of-the-art aeroservoelastic model operating in a variety of different turbulent winds. Tests are conducted in realistic scenarios, in the sense that rebalancing is

performed while the wind turbine is operating in changing wind conditions, including modifications in air density, wind speed, shear, yaw misalignment, upflow angle and turbulence intensity. Details on the specific combinations of conditions used in the tests are reported in appendix A. In addition, §3.4 presents a study assessing the effects of measurement noise on the method performance, with the goal of defining minimum specification requirements for the installed sensors. Finally, section 4 draws conclusions and gives an outlook on future work.

## 2    Methods

### 2.1    Spectral analysis of an unbalanced rotor

In a balanced rotor with $B$ blades, under the assumption of a periodic response, loads transmitted from the rotating frame of reference to the fixed frame contain only $nB$P frequencies. Indeed, the rotor acts as a *filter*: while the full spectrum of frequencies is observed in the rotating frame (1P, 2P, 3P, 4P, ...), in the fixed frame only frequencies that are multiples of the

number of blades do appear ($B$P, $2B$P, $3B$P, ...).

On the other hand, when an imbalance is present, other harmonic components can be detected in fixed frame measurements, the most prominent being typically the 1P harmonic. Hence, detection and correction of rotor imbalances can be based on the analysis of the 1P harmonic measured in the fixed frame.



As an example, consider the measurement of nacelle fore-aft accelerations, which are primarily caused by fluctuations in the rotor thrust. The thrust force $t$ on the rotor can be computed by summing up the out-of-plane shear forces $t_i$ of the $B$ blades, as illustrated in Fig. 1.

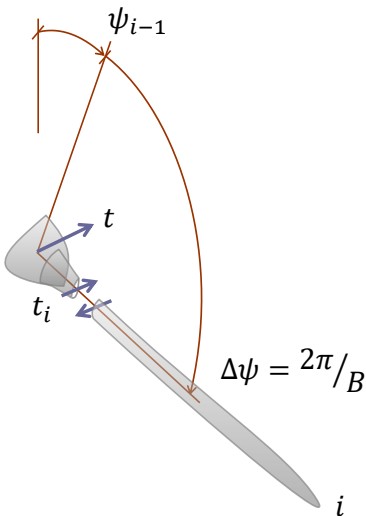

**Figure 1.** Shear force $t$ computed in terms of the shear forces $t_i$ of the $B$ blades. One single blade is shown, for clarity.

The shear force of the generic $i$th blade can be expanded in Fourier series as

$$t_i = t_{0_i} + \sum_{n=1}^{\infty} \left( t_{nc_i} \cos(n\psi_i) + t_{ns_i} \sin(n\psi_i) \right), \tag{1}$$

where $\psi_i = \psi_1 + 2\pi(i-1)/B$ is the azimuthal angle, subscripts $(\cdot)_{nc}$ and $(\cdot)_{ns}$ refer to the $n$P cosine and sine components, respectively, whereas $t_0$ is the 0th harmonic constant amplitude.

Assuming a periodic response, the harmonic amplitudes are the same for the various blades, i.e.

$$t_0 = t_{0_i} = t_{0_j}, \tag{2a}$$

$$t_{nc} = t_{nc_i} = t_{nc_j}, \tag{2b}$$

$$t_{ns} = t_{ns_i} = t_{ns_j}. \tag{2c}$$

In the presence of an imbalance, the harmonic amplitudes of the $k$th (unbalanced) blade will differ from the other ones, and can be expressed as

$$t_{0_k} = t_0 + \delta t_0, \tag{3a}$$

$$t_{nc_k} = t_{nc} + \delta t_{nc}, \tag{3b}$$

$$t_{ns_k} = t_{ns} + \delta t_{ns}. \tag{3c}$$





Inserting now Eqs. (2) and (3) into Eq. (1) and using the properties of trigonometric functions, one can readily compute the thrust force $t$ as

$$t = \sum_{i=1}^{B} t_i, \tag{4a}$$

$$= B t_0 + B \sum_{n=1}^{\infty} (t_{nBc} \cos(nB\psi) + t_{nBs} \sin(nB\psi)) + \delta t_0 + \sum_{n=1}^{\infty} (\delta t_{nc} \cos(n\psi_k) + \delta t_{ns} \sin(n\psi_k)), \tag{4b}$$

where $\psi = \psi_1$. Equation (4b) states that, when the rotor is balanced (i.e., when $\delta t_0 = \delta t_{nc} = \delta t_{ns} = 0$), then only $nB$ harmonics are present in the spectrum of $t$. On the other hand, when the rotor is unbalanced:

1. Also intermediate harmonics pollute the spectrum;

2. The phase of these harmonics indicates the unbalanced blade.

Limiting the analysis to the case of the lowest harmonics of both expansions in Eq. (4b), which are typically the most energetic
ones, leads to

$$t = (B t_0 + \delta t_0)_{0P} + (\delta t_{1c} \cos \psi_k + \delta t_{1s} \sin \psi_k)_{1P} + B (t_{Bc} \cos(B\psi) + t_{Bs} \sin(B\psi))_{BP}. \tag{5}$$

This expression states that the 1P harmonic in the fixed frame is generated by the 1P harmonic of the unbalanced blade. This is not always the case, as the result depends on the considered fixed frame load. For example, similar derivations performed for the nodding (overturning) moment show that in that case also the 0P of the unbalanced blade contributes to the 1P in the fixed
frame. This is beneficial because, as shown later on, the 0 and 1P imbalance harmonics have a different aerodynamic origin. In fact, numerical experiments show that an improved performance and robustness of the detection algorithm can be obtained by using as imbalance detection signal the overturning or yawing moments. However, since load sensors in the fixed frame are typically difficult to install, a similar effect can be obtained by using the difference of two fore-aft accelerometers located in the nacelle at a distance between the two of them (which, depending on their positions, will measure nodding or yawing motions
of the rotor, or combinations thereof).

To better understand the effects of a pitch imbalance, the expression for the aerodynamic contribution to the shear in a blade can be worked out analytically. Following the approach of Manwell et al. (2009), which uses a one degree of freedom rigid body model of a flapping blade, the shear $t_i$ of the $i$th rotor blade is found to be

$$t_i = t_{0_i} + t_{1c_i} \cos \psi_i, \tag{6}$$

where the 0 and 1P harmonic amplitudes write

$$t_{0_i} = -\bar{t} \left( \frac{\Lambda}{2} - \frac{\theta_i}{3} \right), \tag{7a}$$

$$t_{1c_i} = \bar{t} \left( (\Lambda - \theta_i) \bar{V} + \frac{K}{3} \bar{U} \right). \tag{7b}$$





In these expressions, $\bar{t} = \gamma J \Omega^2/(2R)$, $\gamma = \varrho c C_{L,\alpha} R^4/J$ is the Lock number, $\varrho$ the air density, $c$ the blade chord, $C_{L,\alpha}$ the lift slope, $R$ the rotor radius, $J$ the flapping moment of inertia, while $\Lambda = (1-a)U/(\Omega R)$ is the non-dimensional flow velocity at the rotor disk, $a$ being the axial induction, $\bar{V}_0 = V_0/(\Omega R)$ is the non-dimensional cross-flow, and $\bar{U} = U/(\Omega R)$ the non-dimensional wind speed, $\Omega$ being the rotor angular velocity and $K$ the linear vertical wind shear.

Assuming a pitch misalignment $\delta\theta$, the resulting imbalance 0 and 1P harmonic amplitudes are:

$$\delta t_0 = \frac{\bar{t}}{3}\delta\theta, \tag{8a}$$

$$\delta t_{1c} = -\bar{t}\bar{V}\delta\theta. \tag{8b}$$

These expressions state that there is a linear dependency between a pitch misalignment and the resulting harmonic disturbances. In addition, the 1P imbalance harmonic $\delta t_{1c}$ that —according to Eq. (5)— causes the appearance of a 1P harmonic in the fixed

frame, is proportional to the cross-flow. Although in operation there will always be some small misalignment between the rotor axis and the wind vector, this expression suggests that the 1P signal could be strengthened by operating at a slight yaw misalignment with the incoming wind when detecting an imbalance and correcting for it.

A word of caution is due in the interpretation of these analytical results. First of all, this analysis is based on the sole thrust force, while terms other than the cross-flow contribute to the 1P harmonic when considering yawing and nodding moments. In

addition, the model is the simplest possible, using one single degree of freedom and including various simplifications in the derivations. Nonetheless, the model is at least useful in qualitatively understanding the basic mechanisms by which fixed-frame vibrations are caused in an imbalanced rotor. After having served its purpose, the analytical model is dropped from the rest of the paper, whose further developments are not based on it.

### 2.2   Linear imbalance-disturbance model

In this work, an imbalance-disturbance model is assumed in the form

$$\boldsymbol{s} = \boldsymbol{C}(\boldsymbol{b} - \boldsymbol{b}_m), \tag{9a}$$

$$= \boldsymbol{C}\boldsymbol{b} + \boldsymbol{s}_m. \tag{9b}$$

The 1P harmonic amplitude vector of the fixed-frame measured signal $s$ is noted $\boldsymbol{s} = (s_c, s_s)^T$, where $s_c$ and $s_s$ are the cosine and sine components, respectively. Considering here and in the following the common case of a three-bladed rotor ($B = 3$), vector

$\boldsymbol{b} = (b_1, b_2, b_3)^T$ contains the pitch adjustments $b_i$ for each one of the blades, while $\boldsymbol{b}_m$ is the unknown pitch misalignment. Equation (9a) states that, if one knew the misalignment $\boldsymbol{b}_m$, then by pitching the blades by $\boldsymbol{b} = \boldsymbol{b}_m$ one would obtain $\boldsymbol{s} = 0$, i.e. the rotor would be balanced. On the other hand, before rebalancing $\boldsymbol{b} = 0$ and, hence, according to Eq. (9b) one measures a 1P signal equal to $\boldsymbol{s}_m = -\boldsymbol{C}\boldsymbol{b}_m$. In the model, the matrix of coefficients $\boldsymbol{C}$ links imbalance angles and 1P disturbances, and it is defined as

$$\boldsymbol{C} = \begin{bmatrix} c_{c_1} & c_{c_2} & c_{c_3} \\ c_{s_1} & c_{s_2} & c_{s_3} \end{bmatrix}. \tag{10}$$





The model coefficients $C$ and $s_m$ are unknown. However, they can be readily identified from measurements. Once the model is known, one can use it to compute the pitch adjustment $b$ that rebalances the rotor.

Notice that the assumed imbalance-disturbance model implies a linear relationship between the pitch misalignment of the blades and the 1P harmonic component of a measured response signal (acceleration or load) in the fixed frame. As shown later on, this assumption is not a limitation of the model, because in fact the model can be iteratively identified as the rotor is rebalanced, this way effectively removing the linearity hypothesis. However, linearity is confirmed by the previously derived simple analytical model, and it is indeed generally also observed in extensive numerical simulations conducted by using state-of-the-art aeroelastic models.

Since it is nearly impossible to guarantee that the whole model identification and rebalancing procedure will be conducted in exactly the same wind conditions, it is important to reduce the dependency of the model on the operating point. To this end, the harmonic amplitude vector $s$ in Eq. (9) is non-dimensionalized by the dynamic pressure $q = 1/2\varrho U_a^2$, where $U_a$ is a moving-average of the hub-height wind speed, as measured at the nacelle wind vane. This has the effect of making the model coefficients $C$ and $s_m$ largely independent from the operating condition.

To simplify the identification of the model coefficients, the radial symmetry of the rotor can be exploited. Assuming a periodic response, the effects of a misalignment in the second blade will be the same as those caused by a misalignment in the first blade, but shifted by $2\pi/3$. Hence, the model coefficients must obey the following relationship:

$$\left\{ \begin{matrix} c_{c_2} \\ c_{s_2} \end{matrix} \right\} = \left[ \begin{matrix} \cos(2\pi/3) & \sin(2\pi/3) \\ -\sin(2\pi/3) & \cos(2\pi/3) \end{matrix} \right] \left\{ \begin{matrix} c_{c_1} \\ c_{s_1} \end{matrix} \right\} = \boldsymbol{R}\boldsymbol{c}. \tag{11}$$

Clearly, the same argument holds for the relationship between the response of blades two and three. Therefore, matrix $C$ only depends on the two coefficients of vector $c$, and can be written as

$$\boldsymbol{C} = \left[ \begin{matrix} \boldsymbol{c} & \boldsymbol{R}\boldsymbol{c} & \boldsymbol{R}^2\boldsymbol{c} \end{matrix} \right]. \tag{12}$$

It is trivial to observe that this implies also the same relationship between the coefficients of blades three and one, closing the loop.

## 2.3 Model identification

Before computing the pitch adjustments that rebalance the rotor, one needs to identify the unknown coefficients in model (9b). To this end, it is convenient to rewrite the imbalance-disturbance model as follows

$$\boldsymbol{s} = \boldsymbol{C}\boldsymbol{b} + \boldsymbol{s}_m, \tag{13a}$$

$$= \boldsymbol{B}\boldsymbol{c} + \boldsymbol{s}_m. \tag{13b}$$

By simple algebraic derivations, one can readily show that matrix $B$ is a sole function of the pitch adjustment $b$, and writes

$$\boldsymbol{B} = \left[ \begin{matrix} B_{11} & B_{12} \\ -B_{12} & B_{11} \end{matrix} \right], \tag{14}$$



where

$$B_{11} = b_1 + \cos(2\pi/3)b_2 + \cos(4\pi/3)b_3, \tag{15a}$$

$$B_{12} = \sin(2\pi/3)b_2 + \sin(4\pi/3)b_3. \tag{15b}$$

At the beginning of the procedure, one has not yet adjusted the rotor pitch, and hence $\boldsymbol{b} = \boldsymbol{b}^{(1)} = 0$. In this condition, a 1P harmonic equal to $\boldsymbol{s}^{(1)}$ is measured on the machine. Next, the pitch of the blades is changed by a chosen amount $\boldsymbol{b}^{(2)}$. In order not to upset the operating condition of the machine, this arbitrary pitch modification should be characterized by a null collective change. In correspondence to this new condition, one measures a 1P harmonic equal to $\boldsymbol{s}^{(2)}$. Considering the two measurements $\boldsymbol{s}^{(1)}$ and $\boldsymbol{s}^{(2)}$ together, one can write

$$\left\{ \begin{array}{c} \boldsymbol{s}^{(1)} \\ \boldsymbol{s}^{(2)} \end{array} \right\} = \left[ \begin{array}{cc} \boldsymbol{B}^{(1)} & \boldsymbol{I} \\ \boldsymbol{B}^{(2)} & \boldsymbol{I} \end{array} \right] \left\{ \begin{array}{c} \boldsymbol{c} \\ \boldsymbol{s}_m \end{array} \right\}, \tag{16}$$

where $\boldsymbol{B}^{(1)}$ and $\boldsymbol{B}^{(2)}$ indicate matrix (14) evaluated in correspondence of vectors $\boldsymbol{b}^{(1)}$ and $\boldsymbol{b}^{(2)}$, respectively. Inverting this relationship, one readily obtains the unknown coefficients $\boldsymbol{c}$ and $\boldsymbol{s}_m$, which fully characterize model (9b).

### 2.4 Rebalancing

Now that model (9b) has been identified, it can be used to rebalance the rotor. Before doing so, however, one should notice that only imbalances *among* the blades will produce a 1P harmonic in the fixed frame. In fact, a collective rotation of all blades by any given angle will not produce any imbalance, and therefore it cannot be detected by a method based on fixed-frame response signals. This implies that one cannot compute the full pitch adjustment vector $\boldsymbol{b}$, but only a zero-collective adjustment that satisfies the relationship $\sum_{i=1}^{3} b_i = 0$. This is also stated by model (9b), which is in fact a rectangular system of two equations in three unknowns.

By appending the zero-collective constraint to the imbalance-disturbance model, one gets

$$\left\{ \begin{array}{c} \boldsymbol{s} \\ 0 \end{array} \right\} = \left[ \begin{array}{c} \boldsymbol{C} \\ \boldsymbol{1}^T \end{array} \right] \boldsymbol{b} + \left\{ \begin{array}{c} \boldsymbol{s}_m \\ 0 \end{array} \right\}, \tag{17}$$

where $\boldsymbol{1} = (1, 1, 1)^T$. Setting $\boldsymbol{s} = 0$, i.e. requesting a null 1P harmonic response in the fixed frame, one readily computes the necessary pitch adjustments as

$$\boldsymbol{b} = - \left[ \begin{array}{c} \boldsymbol{C} \\ \boldsymbol{1}^T \end{array} \right]^{-1} \left\{ \begin{array}{c} \boldsymbol{s}_m \\ 0 \end{array} \right\}. \tag{18}$$

Blades are now pitched by $\boldsymbol{b}$, as computed by Eq. (18). If, after application of the computed pitch adjustment, a 1P harmonic is still detected in the fixed frame, then this might be an indication of a non-exact linearity between pitch imbalance and fixed-frame harmonic amplitude. In this case, one can iterate the whole procedure. The measured amplitude in the current configuration becomes the new data point in the model identification phase. This data point, together with the one measured





just before adjusting the blade pitch, allows for the identification of a new model. Given the new coefficients, the zero-collective constraint is appended to the model, whose inversion yields the new pitch adjustments. The process is repeated until only a negligible 1P harmonic signal is left in the fixed frame. Figure 2 gives a graphical representation of this algorithmic procedure.

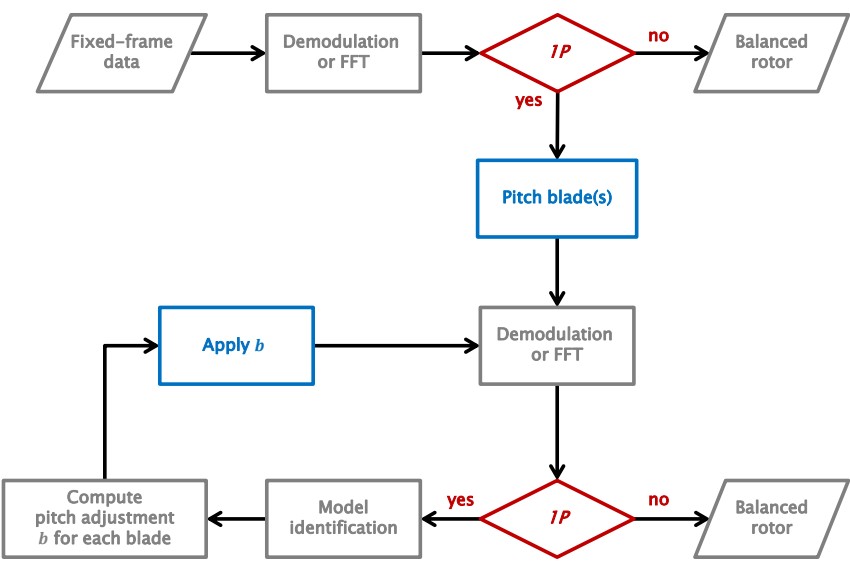

**Figure 2.** Graphical representation of the rotor rebalancing algorithm.

Inspecting the values of the computed pitch adjustments $\boldsymbol{b}$, one may notice in some cases that two blades are characterized
by a same correction, for example $b_1 = b_2 \neq b_3$. This means that only one blade (number 3, in this specific example) was misaligned with respect to the other two. In this case, one might chose to change the blade pitch of blade 3 by $b_3 - 2b_1$, which has the effect of realigning blade 3 with the others instead of adjusting all three at null collective change. This might be useful, for example, in case a blade has been mounted with a wrong pitch offset.

## 3  Results

### 3.1  Simulation environment

In this work, the proposed rebalancing procedure is demonstrated with the help of aeroservoelastic simulations of a 3 MW horizontal axis wind turbine. The machine, characterized by a 80 m hub height and a rotor diameter of 93 m, has cut-in, rated and cut-out speeds equal to 3, 12.5 and 25 ms$^{-1}$, respectively. The transient response of the machine is computed with the finite element multibody code `Cp-Lambda` (Bottasso and Croce, 2006). Rotor blades and tower are modelled using a
15 geometrical exact beam formulation, resulting in a nonlinear finite element model. The rest of the wind turbine is modelled





by a combination of rigid bodies, joints and flexible elements to represent nacelle, drive-train and foundations. Generator and pitch actuators are modelled by first and second order dynamical systems, respectively. The classical blade element momentum theory (BEM) is used to represent the aerodynamics, considering hub and tip-losses, dynamic stall, unsteady aerodynamics and rotor-tower interference. A speed-scheduled linear quadratic regulator (LQR) (Riboldi, 2012) is used for the implementation

of the pitch/torque controller. Turbulent wind time histories of 10 minutes of duration are generated with the code `TurbSim` (Jonkman and Kilcher, 2012), based on the Kaimal turbulence model.

Different combinations of initial pitch misalignments in the range $\pm 2°$ are considered, in which only one, two or even all three blades are simultaneously misaligned. To model finite resolution effects in the pitch system, the minimum resolution of the pitch motion is assumed to be $0.1°$. Therefore, any blade movement smaller than the given resolution is rounded to the

10 closest neighboring integer multiple. To quantify the effectiveness of the rebalancing algorithm, the absolute residual pitch misalignment angle $\epsilon$ is defined as

$$\epsilon = \max(\boldsymbol{b}_m - \boldsymbol{b}) - \min(\boldsymbol{b}_m - \boldsymbol{b}), \tag{19}$$

where $\boldsymbol{b}_m - \boldsymbol{b}$ is the difference between real and computed misalignments.

Accelerometers are placed on the machine main bearing, with the goal of measuring the fixed frame response of the system,

and they are simulated in the mathematical model including the effects of sensor noise. Various tests were conducted in order to identify an optimal accelerometer configuration. Typically, the best results were obtained when two accelerometers are located to the two sides of the main bearing, as spaced as possible from each other. The two accelerometer signals are subtracted one from the other, yielding a differential measurement proportional to the yawing accelerations of the rotor.

## 3.2 Linearity

The model described in §2.2 is based on the assumption that 1P harmonics in the fixed frame depend linearly on the pitch misalignment angle. To validate this assumption, simulations were performed to study the wind turbine fixed frame response to blade misalignments. The simulations were performed in steady sheared wind conditions, misaligning one blade at a time.

Figure 3 shows the sine and cosine differential acceleration components at the main bearing for each one of the three blades. The plots correspond to a wind condition of 7 ms$^{-1}$, although similar results were obtained for different wind speeds. Accel-

25 erations were scaled with respect to the dynamic pressure and averaged over the simulation time. The relationship between 1P response and pitch misalignment appears to be linear to a very good approximation, the correlation coefficient of the linear best fits differing from one by less that $10^{-3}$.

It is interesting to observe that the misalignment of each different blade leaves a unique fingerprint on the measured signal. This means that the linear model not only contains information on the severity of the misalignment, but also on where the

30 misalignment is located.





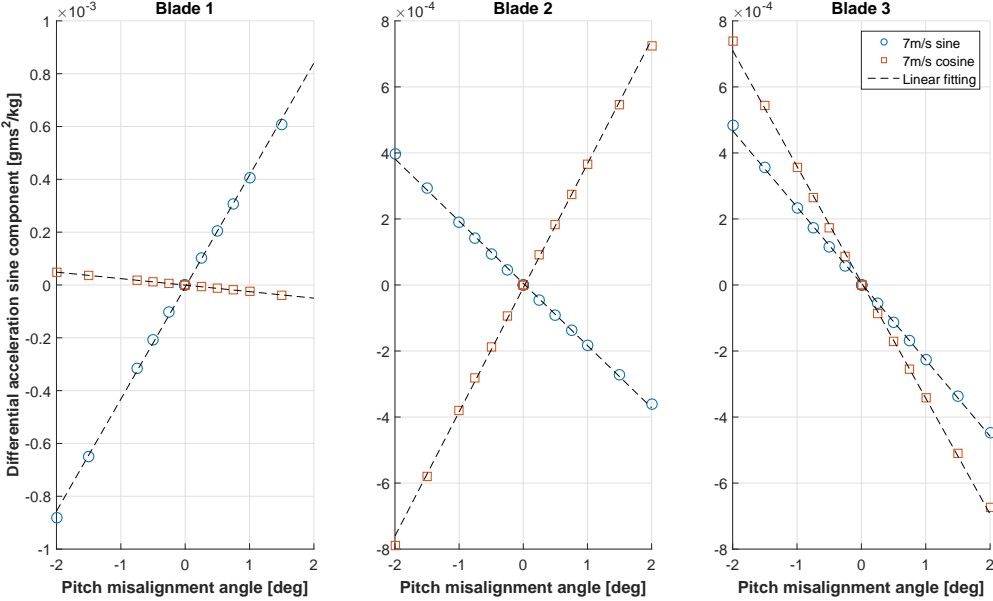

**Figure 3.** Cosine (squares) and sine (circles) 1P components of the main-bearing differential acceleration as functions of pitch misalignment.

### 3.3 Performance assessment of the rebalancing algorithm

Next, the performance of the proposed algorithm is tested in a variety of different wind conditions. The model expressed by Eq. (16) is identified from accelerometer measurements recorded in 10-minute turbulent conditions, characterized by different values of air density, wind speed, turbulence intensity (TI), yaw misalignment, wind shear and upflow. These quantities are assumed to change according to a number of scenarios, termed series A through F, described in detail in appendix A. Once the model is identified, the rotor is rebalanced inverting the model itself. The procedure of identification-rebalancing is then repeated until the residual 1P harmonic is smaller than a given threshold.

Figure 4 shows the absolute residual pitch misalignment $\epsilon$ after each iteration of the rebalancing algorithm. The specific cases reported in the figure correspond to situations where wind speed and TI are kept constant, whereas mean values of yaw misalignment, vertical shear and upflow angle do vary throughout the identification-rebalancing sequence according to what specified for series A through D.

In the figure, the abscissa represents the various steps of the procedure. At the beginning (step 0), a 1P acceleration is measured in the fixed frame. Next, one or more blades are randomly pitched (step 1), while keeping the collective constant. In the resulting new configuration, a new 1P acceleration is measured. Since this step is random, the unbalance of the blades may worsen in this first step. The algorithm is now applied by first identifying the model and then computing the pitch adjustment $\boldsymbol{b}$ that rebalances the rotor. The blades are then accordingly pitched (step 2). If a residual 1P harmonic is still present, the




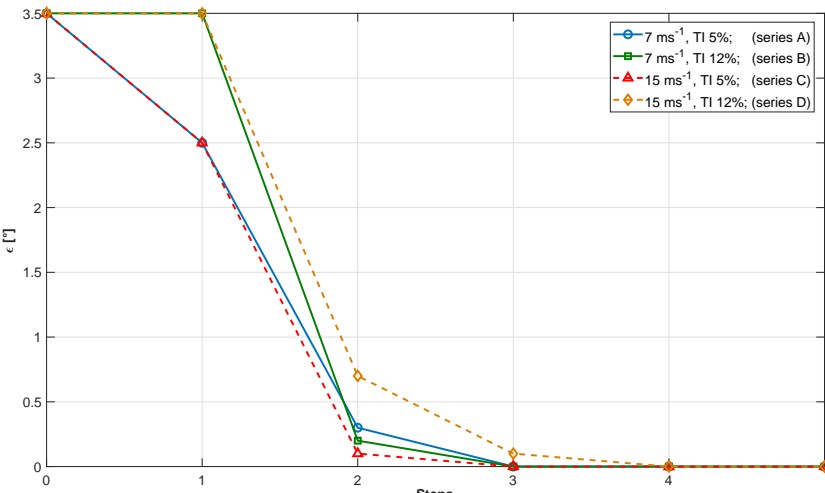

**Figure 4.** Residual pitch misalignment as a function of the number of steps for given wind speed and turbulence intensity, but variable conditions according to series A through D.

algorithm is applied again using data from steps 1 and 2, resulting in a new pitch adjustment (step 3). The procedure is repeated until convergence.

The figure shows that the proposed algorithm is capable of rebalancing the rotor in a very small number of steps, typically ranging between 3 and 4. It should be noticed that during each one of these steps, the machine is operating in markedly different

operating conditions, as described by the series reported in the appendix. Notwithstanding these very significant operational changes, the procedure seems to be quite robust.

An important remark is due at this point. As wind conditions may change from one step to the next, in general it is not possible to guarantee that the imbalance will always diminish at each step of the algorithm. Indeed, some of the following numerical experiments show that the imbalance may occasionally increase. However, this happens only in case of radical changes

in wind conditions from one step to the next. It would be relatively straightforward to avoid such situations by implementing some simple logic in the procedure. For example, one might monitor the operating parameters and continue with rebalancing only when changes do not exceed a certain threshold. In addition, if one observes an increase in the 1P harmonic amplitude after a rebalancing step, then that step might be rejected and the blades could be pitched back to their previous setting. To consider a worst case scenario, in all numerical experiments presented here these simple precautions were not taken. Therefore

the algorithm was forced to continue no matter the severity of operating changes. Because of this, the results show occasional increases of the imbalance throughout the iterations. Nevertheless, these same results also show that the algorithm was always eventually able to successfully rebalance the rotor in a very small number of steps.



Figure 5 reports results obtained at different TI levels for cases characterized by changes in wind speed from 7 to 15 ms$^{-1}$ and in density from 1.225 to 1.1 kgm$^{-3}$ for series E and F. For the E series results, the situation temporarily worsens between steps 1 and 2. This may be due to the simultaneous change of air density, of yaw misalignment and to the halving of shear from 0.4 to 0.2 in this step. Here again, very variable inflow conditions do not seem to excessively affect the performance of

the algorithm, which is indeed able to completely rebalance the turbine rotor within 4 steps.

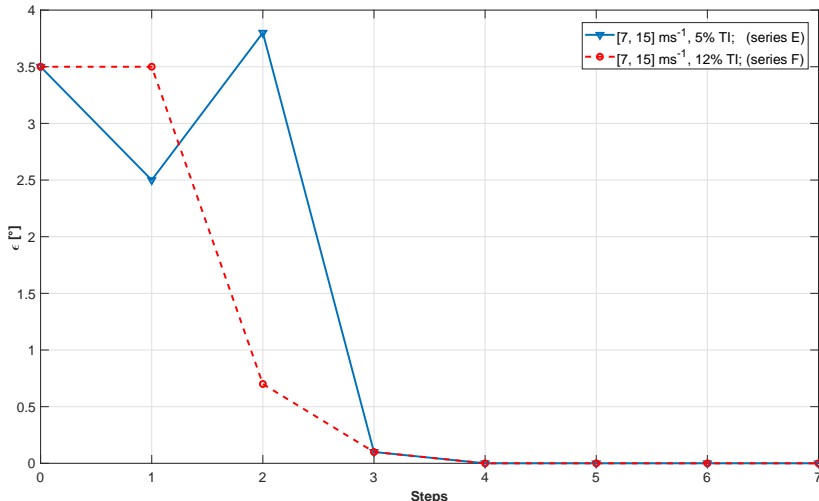

**Figure 5.** Residual pitch misalignment as a function of the number of steps for given turbulence intensity and variable wind speed, but variable conditions according to series E and F.

### 3.4 Effects of sensor noise

The effects of noise on the measurement of the accelerations driving the algorithm was then investigated. In fact, small imbalances induce only small 1P harmonics in the fixed frame, so that the effects of noise on the measurements can be significant.

Sensor noise is modelled by adding a white gaussian signal to the accelerations measured on the multibody wind turbine

model. Five different signal to noise ratios (SNR) are considered, namely SNR=[5 15 22 26 30] dB. To obtain statistically relevant results, for each SNR six different random noise realizations are used, and results are then averaged.

#### 3.4.1 Non-turbulent wind conditions

To separate the effects of sensor noise from the stochastic disturbances caused by turbulence, series composed of 3-minute-long non-turbulent wind conditions are considered first.

Figure 6 shows the average residual pitch misalignment for different SNRs, for a case where all wind parameters are constant and wind speed is equal to 11 ms$^{-1}$. The results clearly illustrate the detrimental effects of decreasing SNR values on the quality





of the rebalancing. For SNR= 5 dB, the residual $\epsilon$ converges to about 0.35°, which is nevertheless a good result considering that in this particular case the initial imbalance was of 1.5°. Increasing SNR, the residual misalignment improves as expected, showing that, from SNR$\geq$ 22 dB and higher, $\epsilon$ converges to values smaller than 0.1° (which is the assumed minimum resolution of the pitch system, and therefore, past this value, differences among the SNR levels become irrelevant).

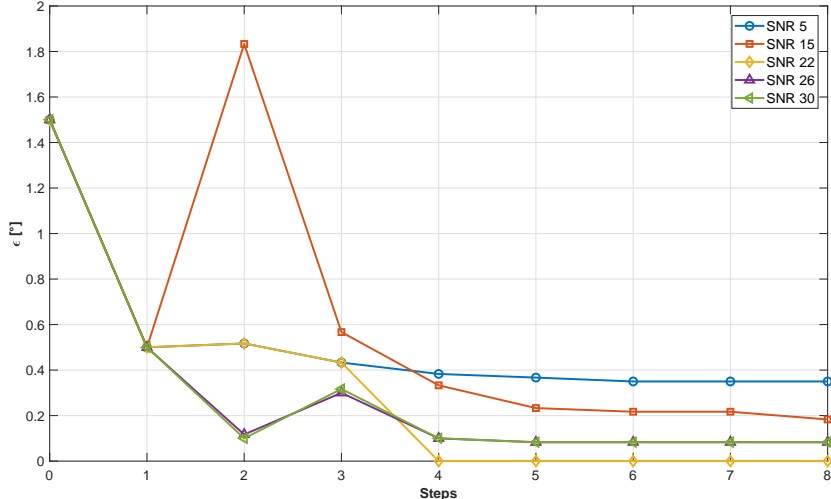

**Figure 6.** Residual pitch misalignment as a function of the number of steps for different SNRs in constant uniform inflow at 11 ms$^{-1}$, $\phi$=0°, $\kappa$=0.4, $\chi$=0°, $\rho$=1.225 kgm$^{-3}$.

5    Figure 7 shows results obtained in varying wind conditions. Specifically, wind speed and density change respectively from 11 to 15 ms$^{-1}$ and from 1.1 to 1.225 kgm$^{-3}$, while vertical shear and misalignment angles vary according to series G. Here again a temporary worsening of the rotor balancing can be observed between step 2 and 3, probably due to the halving of shear between these two steps, accompanied by simultaneous substantial increases in air density and wind speed.

    It appears that the method very effectively reduces the initial misalignments. Indeed, results show a very modest effect

10   of SNR, except for the lowest value of 5 dB that seems to take a bit longer to converge. The apparently surprising lack of sensitivity to SNR can be explained by the changing yaw misalignment within the steps. Indeed, as shown in Eq. (8), the 1P harmonic measured in the fixed frame is related to the presence of a cross flow component. Therefore, a bit of misalignment of the rotor axis with respect to the wind vector eases rebalancing because it makes the effects of an unbalance more prominent, and therefore less affected by noise.





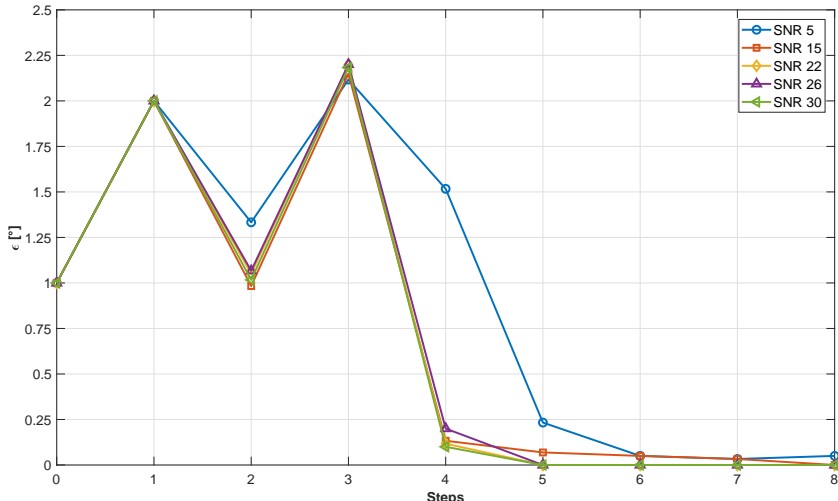

**Figure 7.** Residual pitch misalignment as a function of the number of steps for different SNRs for variable non-turbulent inflow (series G).

### 3.4.2 Turbulent wind conditions

The performance of the method in non-turbulent wind conditions suggests that SNR≥22 dB is sufficient to yield rebalancings within the assumed accuracy of the pitch system. Therefore, this noise value was used as the lower bound for the subsequent analyses.

Figure 8 shows the same simulation series of Fig. 7 (i.e. non-turbulent inflow with wind speed and density changing from 11 to 15 ms$^{-1}$ and from 1.1 to 1.225 kgm$^{-3}$, respectively, with other wind parameters according to series G), but for a turbulent inflow characterized by TI=5%. Here again it appears that SNRs larger than 22 dB have very little effects on the speed of convergence of the algorithm.

    It is also interesting to observe that convergence is actually faster in turbulent conditions (Fig. 8), than in non-turbulent ones
(Fig. 7). This may be due again to the fact that turbulence implies a higher excitation of the 1P harmonic, making it more evident against the sensor noise.

    A large number of tests were performed in additional operating conditions and SNR values, confirm the findings reported herein. Clearly, one should choose a sensor with the highest SNR possible in the frequency range of interest. However, these results suggest that SNR≥30 dB should be typically sufficient for the algorithm to completely rebalance a rotor in turbulent
and varying wind conditions.




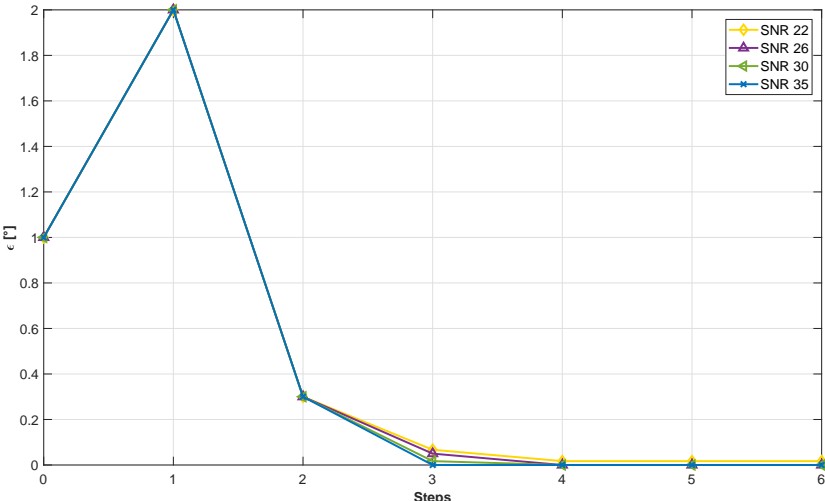

**Figure 8.** Residual pitch misalignment as a function of the number of steps for different SNRs with turbulent inflow at TI=5% (series G).

## 4   Conclusions

This paper has described a new method to detect and correct pitch imbalances in wind turbine rotors. The method uses a measured signal in the fixed frame, typically in the form of accelerations or loads. The signal is demodulated to extract the 1P harmonic, which is then related to the misalignment of the blades by a linear model. By exploiting the axial symmetry of the rotor, the phase of the signal is used to detect which blades are unbalanced. The use of the rotor axial symmetry has the additional effect of reducing the number of free parameters in the model to only two.

The model parameters are readily identified by measuring the signal and computing its harmonics at two different pitch settings, something that is easily achieved by simply pitching the blades by a small chosen amount. The procedure can be performed while the machine is in operation, without shutting it down. The method also works if measurements are taken at different operating conditions, which is indeed inevitable in the field. Once the model has been identified, its inversion readily yields the pitch adjustments of the various blades that rebalance the rotor. If, after rebalancing, some remaining 1P harmonic is detected, the whole procedure can be repeated, thereby eliminating the effects of possible small non-linearities in the imbalance-disturbance relationship. The whole approach has fairly minimal requirements, as it only assumes the availability of a sensor of sufficient accuracy and bandwidth to detect the 1P harmonic to the desired precision, and the ability to command the pitch setting of each blade independently from the others.

Extensive numerical simulations were conducted with the proposed procedure, using a detailed aeroservoelastic model of a multi-MW wind turbine. The analysis considered realistic scenarios, where measurements and rebalancing were performed in operating conditions characterized by varying air density, wind speed, yaw misalignment, upflow, shear and turbulence intensity. The simulation environment also considered the modeling of sensor noise and disturbances.



Based on the results presented herein, the following conclusions may be drawn:

- The relationship between pitch imbalance and 1P fixed frame harmonics appears to be linear and unique depending on the location of the misalignment. This allows one to not only quantify the severity of the imbalance, but also the unbalanced blade(s).

- In realistic wind conditions —i.e. with turbulent wind and variable air density, speed, vertical shear and wind-rotor angles—, the proposed algorithm successfully rebalances the rotor typically within 4 iterations. To account for possible changes in the mean value of wind speed and/or density, the simple scaling of the 1P input by the dynamic pressure was sufficient to guarantee a good performance in all tested conditions.

- Given the relatively small magnitude of the signals that are generated by small misalignments of the blades, particular attention has to be paid to the selection of the installed sensors. Indeed, results have shown that accelerometers with a SNR$\geq$30 dB in the frequency range of interest should be adequate for the present application. However, one should keep in mind that different results might have been obtained on different wind turbines and when placing the sensors at different locations than the ones considered here.

- Good results were obtained by using observation windows of 10 minutes. Although longer time windows might appear to be beneficial to smooth out fluctuations due to turbulence and noise, one should also consider that long time windows might also imply significant changes in rotor speed, which should also be duely accounted for.

Notwithstanding the very promising results obtained here in a simulation environment, a demonstration in the field remains indispensable to prove the actual effectiveness and applicability of the proposed method in practice.

**Nomenclature**

| | |
|---|---|
| $b_i$ | Pitch adjustment that rebalances blade $i$ |
| $b_{m_i}$ | Pitch misalignment of blade $i$ |
| $s$ | Fixed frame signal |
| $t$ | Rotor thrust |
| $t_i$ | Out-of-plane shear of blade $i$ |
| $B$ | Number of blades |
| $U$ | Wind speed |
| $V_0$ | Cross-flow speed |
| | |
| $\theta$ | Pitch angle |
| $\varrho$ | Air density |
| $\psi$ | Azimuthal angle |





| | | |
|---|---|---|
| | $\Omega$ | Rotor angular velocity |
| | $\phi$ | Yaw misalignment angle |
| | $\kappa$ | Vertical shear exponent |
| | $\chi$ | Upflow angle |
| 5 | | |
| | $\boldsymbol{b}$ | Pitch adjustment vector |
| | $\boldsymbol{b}_m$ | Pitch misalignment vector |
| | $\boldsymbol{c}$ | Linear coefficient vector |
| | $\boldsymbol{s}$ | Fixed-frame signal vector |
| 10 | $\boldsymbol{R}$ | Rotation matrix between two consecutive blades |
| | | |
| | $(\cdot)_0$ | Zeroth harmonic |
| | $(\cdot)_n$ | $n$th harmonic |
| | $(\cdot)_i$ | Quantity related to the $i$th blade |
| 15 | $(\cdot)^{(j)}$ | Quantity measured with the $j$th pitch setting |
| | $(\cdot)_c$ | Cosine amplitude |
| | $(\cdot)_s$ | Sine amplitude |
| | $\bar{(\cdot)}$ | Non-dimensional quantity |
| | | |
| 20 | $n$P | $n$ times per revolution |
| | BEM | Blade element momentum |
| | TI | Turbulence intensity |





## Appendix A: Wind Series

The following tables report the values of the relevant operational and wind parameters used for the verification of the rebalancing algorithm.

**Table A1.** Series A. Initial blade misalignment: $\boldsymbol{b}_m = (2°, 0.5°, -1.5°)^T$.

| Step | 0 | 1 | 2 | 3 |
|---|---|---|---|---|
| $U$ [ms$^{-1}$] | 7 | 7 | 7 | 7 |
| TI [%] | 5 | 5 | 5 | 5 |
| $\rho$ [kgm$^{-3}$] | 1.225 | 1.225 | 1.225 | 1.225 |
| $\phi$ [°] | 0 | 10 | 0 | 10 |
| $\kappa$ [-] | 0.2 | 0.4 | 0.2 | 0.2 |
| $\chi$ [°] | 0 | 0 | 0 | 0 |

**Table A2.** Series B. Initial blade misalignment: $\boldsymbol{b}_m = (0.5°, -1.5°, 2°)^T$.

| Step | 0 | 1 | 2 | 3 |
|---|---|---|---|---|
| $U$ [ms$^{-1}$] | 7 | 7 | 7 | 7 |
| TI [%] | 12 | 12 | 12 | 12 |
| $\rho$ [kgm$^{-3}$] | 1.225 | 1.225 | 1.225 | 1.225 |
| $\phi$ [°] | 10 | 10 | 10 | 10 |
| $\kappa$ [-] | 0.4 | 0.4 | 0.4 | 0.2 |
| $\chi$ [°] | 0 | 0 | 0 | 0 |

**Table A3.** Series C. Initial blade misalignment: $\boldsymbol{b}_m = (2°, 0.5°, -1.5°)^T$.

| Step | 0 | 1 | 2 | 3 |
|---|---|---|---|---|
| $U$ [ms$^{-1}$] | 15 | 15 | 15 | 15 |
| TI [%] | 5 | 5 | 5 | 5 |
| $\rho$ [kgm$^{-3}$] | 1.225 | 1.225 | 1.225 | 1.225 |
| $\phi$ [°] | 0 | 0 | 10 | 10 |
| $\kappa$ [-] | 0.2 | 0.2 | 0.4 | 0.2 |
| $\chi$ [°] | 0 | 0 | 0 | 0 |



**Table A4.** Series D. Initial blade misalignment: $\boldsymbol{b}_m = (0.5°, 2°, -1.5°)^T$.

| Step | 0 | 1 | 2 | 3 | 4 |
|---|---|---|---|---|---|
| $U$ [ms$^{-1}$] | 15 | 15 | 15 | 15 | 15 |
| TI [%] | 12 | 12 | 12 | 12 | 12 |
| $\rho$ [kgm$^{-3}$] | 1.225 | 1.225 | 1.225 | 1.225 | 1.225 |
| $\phi$ [°] | 0 | 10 | 0 | 0 | 0 |
| $\kappa$ [-] | 0.2 | 0.4 | 0.2 | 0.2 | 0.2 |
| $\chi$ [°] | 0 | 0 | 0 | 0 | 0 |

**Table A5.** Series E. Initial blade misalignment: $\boldsymbol{b}_m = (2°, 0.5°, -1.5°)^T$.

| Step | 0 | 1 | 2 | 3 | 4 |
|---|---|---|---|---|---|
| $U$ [ms$^{-1}$] | 15 | 7 | 7 | 15 | 15 |
| TI [%] | 5 | 5 | 5 | 5 | 5 |
| $\rho$ [kgm$^{-3}$] | 1.225 | 1.225 | 1.1 | 1.225 | 1.225 |
| $\phi$ [°] | 0 | 10 | 0 | 10 | 0 |
| $\kappa$ [-] | 0.2 | 0.4 | 0.2 | 0.4 | 0.2 |
| $\chi$ [°] | 0 | 0 | 0 | 0 | 0 |

**Table A6.** Series F. Initial blade misalignment: $\boldsymbol{b}_m = (1°, 2°, -1.5°)^T$.

| Step | 0 | 1 | 2 | 3 | 4 |
|---|---|---|---|---|---|
| $U$ [ms$^{-1}$] | 15 | 7 | 7 | 15 | 15 |
| TI [%] | 12 | 12 | 12 | 12 | 12 |
| $\rho$ [kgm$^{-3}$] | 1.225 | 1.225 | 1.1 | 1.225 | 1.225 |
| $\phi$ [°] | 10 | 10 | 0 | 10 | 0 |
| $\kappa$ [-] | 0.4 | 0.4 | 0.2 | 0.4 | 0.2 |
| $\chi$ [°] | 0 | 0 | 0 | 0 | 0 |

**Table A7.** Series G. Initial blade misalignment: $\boldsymbol{b}_m = (-1°, 0°, 0°)^T$.

| Step | 0 | 1 | 2 | 3 | 4 | 5 | 6 | 7 | 8 |
|---|---|---|---|---|---|---|---|---|---|
| $U$ [ms$^{-1}$] | 15 | 11 | 11 | 15 | 11 | 15 | 15 | 11 | 15 |
| $\rho$ [kgm$^{-3}$] | 1.1 | 1.225 | 1.1 | 1.225 | 1.225 | 1.1 | 1.225 | 1.1 | 1.1 |
| $\phi$ [°] | 0 | 0 | 10 | 10 | 0 | 10 | 0 | 10 | 10 |
| $\kappa$ [-] | 0.4 | 0.2 | 0.4 | 0.2 | 0.2 | 0.2 | 0.4 | 0.4 | 0.2 |
| $\chi$ [°] | -4 | 0 | 0 | 0 | 0 | 0 | -4 | 0 | 0 |



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
