# Peer review of "Automatic detection and correction of pitch misalignment in wind turbine rotors"

_Wind Energy Science, 2018_

## Referee Comment (RC1) · Anonymous Referee #1 · 26 Apr 2018

Page 1, Abstract, line 8, "A part from the availability..." do you mean "Apart from the availability..."?

Page 3, Section 1, somewhere in the introduction section, I am wondering how you can distinguish 1P loading due to yaw misalignment from 1P loading due to pitch calibration error? Do you need to assume the yaw misalignment is perfect? How would you address this? I am not sure if I saw this clearly explained in the paper.

Page 10, section 3.2, somewhere in the Linearity section I thought of the issue of what about when the physical blades are not balanced (e.g. some sort of mass discrepancy due to a manufacturing tolerance issue)? Do you need to assume the blades are perfectly identical? I think this should be clarified somewhere in the paper.

[Figure]

Otherwise, a great paper, good work!

---

## Referee Comment (RC2) · Anonymous Referee #2 · 31 May 2018

Thank you for this good and extensive research work on a topic which deals with a isuue which many turbines suffer.

General: The paper is basically very well structured, straight-forward and written profoundly starting with the nice analytical part and having an extensive sensitivity study. The blade angle misalignment is a very important topic for wind turbine load and CoE reduction as well as lifetime extension. Since a large share of turbines is affected, developing such a method is very suitable in order to continuously monitor and adjust the blades. Using the torsional amplitude derived from accelerometers as reliable indicator for blade angle misalignment and correction has been applied successfully at hundreds of turbines in diagnostic offline measurements. Many CMS companies offer wind turbine structural monitoring or even lifetime consumption monitoring where it is

possible that suitable sensors are available. One wind turbine manufacture performs a step-wise pitch rebalancing procedure during operation. which seems quite similar but perhaps with other algorithms. From this point, the successful implementation of the developed method in the field is very it is promising. The simulation-based sensitivity studies are very extensive, and in the conclusions, points to be checked during validation by measurement for field implementation are carefully mentioned.

However, several points need a comment, clarification and/or correction. If some of the mentioned topics here are already covered by another paper it would be beneficent to give more notes in the paper, add references or else. It is clear that many following comments are with a strong relation to experience from field measurements and some consider perhaps the next steps of the presented research.

Introduction, page 1 ff The description of the state of art is incomplete. However it is clear that the introduction has to be short and precise and is not the core part of the work. So, it is up to the editor and authors how many changes are done.

There are no references to published blade angle deviation statistics from field measurements, which have more direct link to the research motivation as statistics on the pitch system failures, .e.g. Heilmann DEWEK 2015 and WindEurope Summit 2016. The mentioned measuring methods are incomplete.

Methods are mentioned which have at present a questionable accuracy (video measurements or drones, please provide references) are mentioned while methods applied for years like laser-based methods, and vibration measurements for validation of accurate blade angle correction are missing (Wobben Patent 2001; Donth EWEA Offshore 2011; Mayer DEWEK 2015; Bartholomay DEWEK2015; Heilmann DEWEK 2015 and WindEurope Summit 2016) . The suitability of the torsional 1P vibration from accelerometers for blade angle misalignment assessment is as well stated.

VDI guideline VDI3834-1:2015 contains an annex on in-situ balancing on wind turbines. Here mass imbalance, blade angle deviation and their combination is discussed and

recommendations for measurement are given with consideration of DIN ISO 21940-13. Moreover, the guideline states requirements for measurements on low frequent turbine vibration measurements and that the measuring device including the evaluation has to meet the requirements, not only the sensor. This has a strong relation to the conclusions of the SNR study in the reviewed paper where the SNR requirement is stated only for the acceleration sensor. This statement needs to be corrected in the paper's conclusions, etc. There is a patented load reduction system implemented by General Electrics in their turbines already for several years for blade angle misalignment determination and correction by temporarily changing the blade angles during operation (exact patent number currently unknown, sorry). The hub movement relative to the fixed frame is measured at 4x90°, so it is not based on accelerometers. This patent should be mentioned, (at least in future papers) and discussed in how far the presented work does not interfere with this patent/ proceÂňdure/ algorithm, etc. If this point is covered already by one of the references of the paper please indicate this.

References with relation to the state of art to document the relevance of the comments, mostly non-scientific, not reviewed papers

Bartholomay et al.: A new approach to elimination of aerodynamic imbalance of wind turbines, Proceedings: DEWEK 2015, Bremen, Germany, 19-20 May, 2015 (test pitch method using axial vibration for refined blade angle correction, perhaps now patented for offline measurements)

Donth, A. et al.: New Methods for Balancing Offshore Wind Turbine Rotors including Blade Angle Measurements. Proceedings: EWEA Offshore 2011, Amsterdam, The Netherlands, November, 22 – 24, 2011 (laser-based blade angle measurement and necessity of validation by vibration measurement)

Donth, A. et al.: Payback Analysis of Different Rotor Balancing Strategies, EWEA 2013,Vienna, Austria, 4-7 February, 2013 (statistics on mass imbalance and blade angl deviaitons, 45% of serial WT in the field are affected by mass and/or aerodyn.

Imbalance)

Heilmann, C. et al.: Field studies on absolute blade angle at wind turbine rotors and their impact on lifetime consumption and yield, Proceedings: DEWEK 2015, Bremen, Germany, 19-20 May, 2015 (blade angles of 277 WT, necessity of vibration measurement for adjustment validation)

Heilmann, C. et al.: Improved Quality of Rotor Balancing Measurements through Criteria from the Guideline VDI-3834-1:2015, WindEurope-Summit 2016, Hamburg, Germany, 27-29 September 2016 (blade angles of 383 WT, non-linearity of pitch angle changes, suitability of torsional signal)

Kleinselbeck et al.: 3D laser optical measurement of the rotor blade angle, Proceedings: DEWEK 2015, Bremen, Germany, 19-20 May, 2015(blade angle measurements using tachymeter)

VDI guideline VDI3834-1:2015: Measurement and evaluation of the mechanical vibration of wind turbines and their components

DE10032314C1 (2001), US7052232 (2006) Wobben (i.e. company Enercon) patent on laser-based blade angle method

Informative: Christoph Lucks (Windcomp GmbH) and other various international patents on laser-based blade angle measuring methods since 2007, DEWEK2015 only the poster: Mayer, Lucks: Laser Based Geometry Measurement of Rotor Blades

Section 2.1, p4, Fig 1: In the text above the thrust force t is introduced. However, fig. 1 contains the name changed to shear force. This has to be consistent. Page 5, 6 , especially eq. 7 and 8: many symbols not contained in nomenclature, see last page.

Results Section 3.1. Simulation environment and 3.2 Linearity: Page 7, line 9 to 13 and section 3.1. Simulation environment and 3.2 Linearity: (If some of the comments are covered with other references please indicate this). Obtaining independence from operating point using dynamic pressure from nacelle anemometer Please

short explanation why well-known non-linearities from tower resonance during imbalance measurements (e.g. Gasch/Twele "Wind Power Plants", Springer 2005, chapter 8; VDI3834-1:2015) which affect the axial acceleration measurement and nacelle anemometer behaviour can be removed by normalizing the dynamic pressure. Please state for the simulation environment turbine rotor speed range and natural frequencies of tower simulated to show that resonance effects are. Many towers have (1st bending) resonance frequency within the operating range, resonance affects axial and lateral measurement. Lattice towers may have problems with resonance from torsional vibration. It is well known that the nacelle anemometer cannot be trusted so much. It is affected by the rotor wake, its position on the nacelle etc. The calibration of the nacelle anemometer transfer function is a big point of discussion in the wind industry. WindGuard published a report in 2018 ("Anemometer calibration at different air temperatures and air pressures", https://www.windguard.com/publications-wind-energy-statistics.html) detected significantly different anemometer behaviour under changing climatic conditions. Please explain nacelle anemometer position /measuring point during simulation. Why is it possible to neglect this? Or is it assumed that the turbine control has a very exact nacelle anemometer calibration, then this needs to be stated as requirement like the SNR of the acceleration sensor? These approaches need explanation or stricter requirement statements in the paper's results/conclusions section.

The equation of the dynamic pressure needs formatting and numbering like other equations, despite it is perhaps trivial for the authors.

There should be the equation for normalisation of acceleration with the dynamic pressure and a related symbol introduced and used in fig. 3 etc. to prevent misunderstandings. The averaging interval of $U_a$ is not mentioned.

Figure 3 needs corrections: Physical property is "Scaled acceleration" which can be found in the text page 10, line 24-24 in a verbal description, and derived from the unit if it comes clear that Us and metric units are mixed, US unit "g" for acceleration and kg

in one single expression. Please use only SI units.

Fig 3 should allow a plausibility check of performed simulation with field measurements (e.g. exemplary values given in the above mentioned papers) and show that the 1P-torsional amplitudes are in the right order of magnitude and the SNR of evaluation. There are no direct 1P- amplitudes given in fig. 3, only the sine and cosine component, hence the following estimation and comparison with the field is only valid if the way of estimation is correct. Estimating the acceleration backwards with the dynamic pressure from Fig. 3 gives e.g. for the sine component for blade 1 at -2° approx. 26 mg. In the field, measured torsional amplitudes from acceleration are in the range below 3 mg for 2° blade angle deviation. For the fine-tuning range below 0.5° they are in the range below 0.5 mg, often below 0.1 mg. At this amplitude level, it is in the noisy measuring environment, especially close to the main bearing in the field very difficult to fulfil the required SNR of 22dB even for high quality sensors. Despite the shown linearity in Fig 3, there are some more questions whether the linearity can be really assumed as well in the field. Is yawing excluded during the performed simulations? Then this is an additional requirement to be mentioned as well in the Conclusions. Is the axial displacement of the nacelle from thrust changes linked with wind speed changes considered during the simulations? Due to the coupling of axial, lateral and torsional vibration this is a very complex topic and may impact on the centre of the torsional vibration, i.e. disturbing the assumed linearity.

The sensitivity studies are very good and extensive. However, the line for SNR5 should be shown in Fig. 8 because a SNR 5 to 10 dB is to be expected in the field, especially for the steps of refined blade angle correction because of various noise sources in the nacelle, especially close to a bearing.

Conclusions, page 16 ff.

It is very good that the step-wise approach promises to be able to cope with non-linearities during the pitch angle correction procedure, which are observed in the field.

It has to be corrected that the SNR requirement is for the entire measuring device including evaluation, not only for the sensors, see above. The paper states the measurement can be performed during normal operation. Many modern turbines have several vibration control / torque control algorithms implemented. How would the proposed method be affected, or is it an additional requirement to be mentioned to turn all this off? Using order analysis requires a trigger to obtain the phase angle correctly, i.e. a rotor speed sensor. So not only the nacelle wind speed and accelerometers but such a signal is required. Despite in the presented simulation the results for 10 min time intervals are consistent, it has to be kept in mind that VDI 3834-1:2015 recommends larger measurement intervals than 10 min for diagnostic measurements. However, as the method uses step by step measurements, the validation of the results is included. Comment for future work: Will there be a sensitivity analysis of the impact of mass imbalance on the proposed method? It is known from the field statistics that at a rotor, mass imbalance and aerodynamic imbalance is often simultaneously present. Mass imbalance limits may be as small as 60 kgm while detected mass imbalances amount up to more than 5000 kgm even today. Mass imbalance causes as well torsional vibration due to its lever to the center of torsional rotation and the fore-aft movement for the tilted rotor.

Page 17: The Nomenclature is incomplete: a, lambda, epsilon, CL, J, R, K, C, (.)a, (.)m, sm

---

## Author Comment (AC1) · 11 Sep 2018

**Reply to Reviewers**

We thank the reviewers for their detailed analysis and constructive inputs. A list of point-by-point replies to the reviewers' comments is detailed in the following.

**Reviewer 1**

1. **Reviewer**: *Page 1, Abstract, line 8, "A part from the availability..." do you mean "Apart from the availability..."?*
   **Authors**:  This was fixed.

2. **Reviewer**: *Page 3, Section 1, "somewhere in the introduction section, I am wondering how you can distinguish 1P loading due to yaw misalignment from 1P loading due to pitch calibration error? Do you need to assume the yaw misalignment is perfect? How would you address this? I am not sure if I saw this clearly explained in the paper."*
   **Authors**: Yaw misalignment will cause 1P loading on the blade, but 3P loading in the fixed system. On the other hand, pitch misalignment will cause 1P loading in the fixed system. From this point of view, the two effects can be clearly distinguished. This is explained in section 2.1 of the paper.

3. **Reviewer**: *Page 10, section 3.2, somewhere in the Linearity section I thought of the issue of what about when the physical blades are not balanced (e.g. some sort of mass discrepancy due to a manufacturing tolerance issue)? Do you need to assume the blades are perfectly identical? I think this should be clarified somewhere in the paper.*
   **Authors**: Please see reply 13 to reviewer 2.

**Reviewer 2**

1. **Reviewer**: *The description of the state of art is incomplete. However it is clear that the introduction has to be short and precise and is not the core part of the work. So, it is up to the editor and authors how many changes are done.*
   **Authors**: The additional references suggested by the reviewer were included in the bibliography, and the introduction modified accordingly.

2. **Reviewer**: *There is a patented load reduction system implemented by General Electrics in their turbines already for several years for blade angle misalignment determination and correction by temporarily changing the blade angles during operation (exact patent number currently unknown, sorry). The hub movement relative to the fixed frame is measured at 4x90° , so it is not based on accelerometers. This patent should be mentioned, (at least in future papers) and discussed in how far the presented work does not interfere with this patent/ procedure/ algorithm, etc. If this point is covered already by one of the references of the paper please indicate this.*
   **Authors**: Based on the reviewer's hint, we identified two patents from GE –of which we were not previously aware– on the topic of wind turbine rotor rebalancing: US 2009/0035136A1 and US 8,683,688 B2. It is true that even in the case of these patents, as in our paper, pitch angle changes are used to identify the source of the imbalance. However, none of the patents provides the mathematical formulation of the method, nor it demonstrates that rebalancing by pitch offsetting is a solvable and well-posed problem. In addition, measurements are assumed to be made on the

shaft, which is a significant difference with our method. In fact, our rebalancing algorithm can in principle be used with different sensors (strain-gauges, accelerometers, …) located in different positions, without modifications to the algorithm itself. Furthermore, our formulation shows that one can rebalance a completely generic case characterized by multiple unbalanced blades. This problem is not even mentioned in the two patents. For completeness, the patents have been added to the bibliography and duly cited in the text.

3. **Reviewer**: *Consistency of definition of thrust: either call it shear or thrust force Section 2.1, p4, Fig 1: In the text above the thrust force t is introduced. However, fig. 1 contains the name changed to shear force. This has to be consistent. Page 5, 6 , especially eq. 7 and 8: many symbols not contained in nomenclature, see last page.*
   **Authors**: The legend of Figure 1 contained an error, and it should read "Thrust force t …" instead of "Shear force t …". This was corrected. Other missing symbols have now been included in the nomenclature.

4. **Reviewer**: *Obtaining independence from operating point using dynamic pressure from nacelle anemometer Please short explanation why well-known non-linearities from tower resonance during imbalance measurements (e.g. Gasch/Twele "Wind Power Plants", Springer 2005, chapter 8; VDI3834-1:2015) which affect the axial acceleration measurement and nacelle anemometer behaviour can be removed by normalizing the dynamic pressure. Please state for the simulation environment turbine rotor speed range and natural frequencies of tower simulated to show that resonance effects are. Many towers have (1st bending) resonance frequency within the operating range, resonance affects axial and lateral measurement. Lattice towers may have problems with resonance from torsional vibration.*
   **Authors**: The text has been updated to address the points raised by the reviewer. In short, the method only uses 1P harmonics, which are typically separated from other frequencies. In addition, one can simply avoid performing the rebalancing at operating points characterized by resonances, since these are typically known with good accuracy.
   In addition, a possible resonance between 1p and tower bending modes will produce fore-aft and side-side vibrations, with only limited effects on the pure yawing and nodding accelerations, which are mainly excited by the tower torsion mode at a higher frequency. By using the differential signal between two accelerometers suitably located in the nacelle, as described at the end of section 3.1, our implementation reduces the effects of fore-aft and side-side motions (affected by the resonance).

5. **Reviewer**: *It is well known that the nacelle anemometer cannot be trusted so much. It is affected by the rotor wake, its position on the nacelle etc. The calibration of the nacelle anemometer transfer function is a big point of discussion in the wind industry. WindGuard published a report in 2018 ("Anemometer calibration at different air temperatures and air pressures", https://www.windguard.com/publicationswind-energy-statistics.html) detected significantly different anemometer behaviour under changing climatic conditions. Please explain nacelle anemometer position /measuring point during simulation. Why is it possible to neglect this? Or is it assumed that the turbine control has a very exact nacelle anemometer calibration, then this needs to be stated as requirement like the SNR of the acceleration sensor? These approaches need explanation or stricter requirement statements in the paper's results/conclusions section.*

**Authors**: The reviewer is certainly right in pointing out the low accuracy of nacelle anemometers. Following the reviewer's suggestion, a sensitivity analysis was performed to study the effect of errors in the anemometer measurements, by assuming errors up to $\pm20\%$ in the wind speed estimates. The effects of such errors were always minimal, leading only to a slight increase in the number of required iterations. This was duly noted in the revised text. In addition, it should also be remarked that one could use a rotor-effective measurement of the wind speed, based for example on the torque-balance equation, this way by-passing the need of relying on the nacelle anemometer. This fact was also duly noted in the revised text, with the addition of a relevant reference.

**Reviewer**: *The equation of the dynamic pressure needs formatting and numbering like other equations, despite it is perhaps trivial for the authors. There should be the equation for normalisation of acceleration with the dynamic pressure and a related symbol introduced and used in fig. 3 etc. to prevent misunderstandings. The averaging interval of Ua is not mentioned.*

**Authors**: The equation has been numbered as requested, and the averaging interval of Ua has been clarified in the text.

**Reviewer**: *Figure 3 needs corrections: Physical property is "Scaled acceleration" which can be found in the text page 10, line 24-24 in a verbal description, and derived from the unit if it comes clear that Us and metric units are mixed, US unit "g" for acceleration and kg in one single expression. Please use only SI units.*

**Authors**: The revised version of the paper now only uses SI units.

6. **Reviewer**: *Fig 3 should allow a plausibility check of performed simulation with field measurements (e.g. exemplary values given in the above mentioned papers) and show that the 1Ptorsional amplitudes are in the right order of magnitude and the SNR of evaluation. There are no direct 1P-amplitudes given in fig. 3, only the sine and cosine component, hence the following estimation and comparison with the field is only valid if the way of estimation is correct. Estimating the acceleration backwards with the dynamic pressure from Fig. 3 gives e.g. for the sine component for blade 1 at -2° approx. 26 mg. In the field, measured torsional amplitudes from acceleration are in the range below 3 mg for 2° blade angle deviation. For the fine-tuning range below 0.5° they are in the range below 0.5 mg, often below 0.1 mg. At this amplitude level, it is in the noisy measuring environment, especially close to the main bearing in the field very difficult to fulfil the required SNR of 22dB even for high quality sensors. Despite the shown linearity in Fig 3, there are some more questions whether the linearity can be really assumed as well in the field.*

**Authors**: Following the reviewer's input, we discovered an unfortunate mistake in the data reported in Figure 3: the magnitude of the "torsional" accelerations should be indeed about 0.9 mg at 7 m/s for a deviation of 2°. The figure was corrected accordingly, and such a mistake was only confined to the plot. Non-linearities can be corrected as discussed in the paper using the proposed method in an iterative fashion. Regarding SNR, please see reply no. 8.

7. **Reviewer**: *Is yawing excluded during the performed simulations? Then this is an additional requirement to be mentioned as well in the Conclusions. Is the axial displacement of the nacelle from thrust changes linked with wind speed changes considered during the simulations? Due to the coupling of axial, lateral and torsional vibration this is a very complex topic and may impact on the centre of the torsional vibration, i.e. disturbing the assumed linearity.*

**Authors**: The simulations were performed under different inflow conditions, and include different yaw misalignment angles, different upflows and vertical shears. Moreover, the aeroelastic model comprises about 2500 degrees of freedom, and includes all possible needed couplings. This was clarified in the text.

8.  **Reviewer**: *The sensitivity studies are very good and extensive. However, the line for SNR5 should be shown in Fig. 8 because a SNR 5 to 10 dB is to be expected in the field, especially for the steps of refined blade angle correction because of various noise sources in the nacelle, especially close to a bearing.*
    **Authors**: Additional results with lower values of SNR were added, and the text updated accordingly. It appears that results are fairly robust with respect to this parameter, and even modest values of SNR allow for a precise rebalancing of the rotor.

9.  **Reviewer**: *It has to be corrected that the SNR requirement is for the entire measuring device including evaluation, not only for the sensors, see above. [..] Moreover, the guideline states requirements for measurements on low frequent turbine vibration measurements and that the measuring device including the evaluation has to meet the requirements, not only the sensor.*
    **Authors**: This is correct, and the text was updated accordingly.

10. **Reviewer**: *The paper states the measurement can be performed during normal operation. Many modern turbines have several vibration control / torque control algorithms implemented. How would the proposed method be affected, or is it an additional requirement to be mentioned to turn all this off?*
    **Authors**: This is a good point, although we think that the answer will depend on the specifics of the control algorithm. In any case, a comment in this sense was added to the text.

11. **Reviewer**: *Using order analysis requires a trigger to obtain the phase angle correctly, i.e. a rotor speed sensor. So not only the nacelle wind speed and accelerometers but such a signal is required.*
    **Authors**: Correct, we assume that rotor speed and azimuth are available, and this is stated in the paper.

12. **Reviewer**: *Despite in the presented simulation the results for 10 min time intervals are consistent, it has to be kept in mind that VDI 3834-1:2015 recommends larger measurement intervals than 10 min for diagnostic measurements. However, as the method uses step by step measurements, the validation of the results is included.*
    **Authors**: The 10-minute window was chosen as a compromise: short windows might not be able to eliminate turbulent effects, while using long windows one might encounter significant changes in operating conditions (e.g., wind speed, direction, etc.). The definition of best practices to be used in the field can only be addressed once the method is comprehensively tested on real turbines.

13. **Reviewer**: *Comment for future work: Will there be a sensitivity analysis of the impact of mass imbalance on the proposed method? It is known from the field statistics that at a rotor, mass imbalance and aerodynamic imbalance is often simultaneously present. Mass imbalance limits may be as small as 60 kgm while detected mass imbalances amount up to more than 5000 kgm*

*even today. Mass imbalance causes as well torsional vibration due to its lever to the center of torsional rotation and the fore-aft movement for the tilted rotor.*

**Authors**: We agree, and in addition there is a need to develop reliable ways of distinguishing these two effects. A note in this sense was added to the conclusions.

14. **Reviewer**: *Page 17: The Nomenclature is incomplete*
    **Authors**: This has now been fixed.

We have taken the opportunity to make several small editorial changes to the text, in order to improve readability. A revised version of the manuscript is attached to the present reply, with the main changes highlighted in red.

Best regards.
The authors

---

## Author Comment (AC2) · 11 Sep 2018

[revised manuscript text omitted]

$$= Bt_0 + B \sum_{n=1}^{\infty} (t_{nB\mathrm{c}} \cos(nB\psi) + t_{nB\mathrm{s}} \sin(nB\psi)) + \delta t_0 + \sum_{n=1}^{\infty} (\delta t_{nc} \cos(n\psi_k) + \delta t_{ns} \sin(n\psi_k)), \tag{4b}$$

where $\psi = \psi_1$. Equation (4b) states that, when the rotor is balanced (i.e., when $\delta t_0 = \delta t_{nc} = \delta t_{ns} = 0$), then only $nB$ harmonics are present in the spectrum of $t$. On the other hand, when the rotor is unbalanced:

1. Also intermediate harmonics pollute the spectrum;

2. The phase of these harmonics indicates the unbalanced blade.

Limiting the analysis to the case of the lowest harmonics of both expansions in Eq. (4b), which are typically the most energetic ones, leads to

$$t = (Bt_0 + \delta t_0)_{0\mathrm{P}} + (\delta t_{1\mathrm{c}} \cos \psi_k + \delta t_{1\mathrm{s}} \sin \psi_k)_{1\mathrm{P}} + B (t_{B\mathrm{c}} \cos(B\psi) + t_{B\mathrm{s}} \sin(B\psi))_{B\mathrm{P}}. \
[revised manuscript text omitted]

---

## Author Response (AR2)

Labels have been enlarged for figures 3 through 8, as requested.